# The Greek Jefferson Scale of Empathy—Medical Student Version (JSE-S): Psychometric Properties and Its Associated Factors

**DOI:** 10.3390/bs14030195

**Published:** 2024-02-28

**Authors:** Polychronis Voultsos, Petros Galanis, Marianna-Foteini A. Dafni, Venetia-Sofia Velonaki, Georgia-Neta Andreou, Leda Kovatsi

**Affiliations:** 1Department of Forensic and Toxicology, Division: Medical Law and Ethics, School of Medicine, Aristotle University of Thessaloniki, 54 124 Thessaloniki, Greece; kovatsi@auth.gr; 2School of Nursing, National and Kapodistrian University of Athens, Goudi, 11 517 Athens, Greece; pegalan@nurs.uoa.gr (P.G.); venvel@nurs.uoa.gr (V.-S.V.); 3Postgraduate Program in Bioethics, School of Medicine, Democritus University of Thrace, 68 100 Alexandroupolis, Greece; mardaf99@gmail.com; 4Michael Cacoyannis Foundation, Cultural Center, 206 Peraeus Str., Tavros, 17 778 Athens, Greece; georgia.andreou11@gmail.com

**Keywords:** empathy, Jefferson Scale of Empathy—Student version (JSE-S), undergraduate medical students, gender, academic year, specialty preferences, belief in God/supreme power

## Abstract

The present study aimed to evaluate the psychometric properties of the Greek version of the Jefferson Scale of Empathy—Student version (JSE-S) and its association with potential predictors among Greek-speaking undergraduate medical students. This study adopted a cross-sectional, comparative–descriptive research design. The study was conducted during October and November 2023. Cronbach’s α values for the JSE-S and the factors “perspective taking”, “compassionate care”, and “standing in the patient’s shoes” showed internal consistency. The intraclass correlation coefficient for the JSE-S score in the test–retest study indicated a high level of reliability. The participants showed moderate empathy levels. Females scored higher than males in the Greek version of the JSE-S. Moreover, students enrolled in the fourth academic year showed higher empathy mean scores than those enrolled in the first year. Statistically significant empathy differences by specialty preferences or faith in God/supreme power were not found. The present study provided satisfactory evidence that the Greek JSE-S is a psychometrically sound measurement instrument. Empathy differences by gender were found in line with prior literature.

## 1. Introduction

Empathy is an integral component of overall clinical competence and the provision of high-quality patient care. Empathy is an indispensable quality in a practicing physician, according to the principles of good medical practice. It is a pillar of medical professionalism [1]. Empathic engagement is a critical component of an effective therapeutic relationship. The physician must be empathetic in implementing the patient-centered model of care, under which a physician should focus on the patient’s perceived physical and psycho-social needs. Physician empathy not only leads to better patient care [2] but is also beneficial to physicians and other healthcare professionals themselves [3].

Physician empathy has been shown to have many beneficial effects on both patients and physicians, and their relationship as well. Physician empathy can help build trust in the patient–physician relationship, enhance patient–physician communication, prevent medical paternalism, improve patient satisfaction and health outcomes, and “support patient self-efficacy, treatment adherence, and disease management planning” [4,5,6,7]. Furthermore, physician empathy enhances the emotional wellbeing of physicians [4]. Robinson et al. state that “studies suggest that healthcare worker empathy is associated with lower malpractice claims rates, job satisfaction, and reduced burnout” [4].

In general terms, empathy is “the process of coming to know, understand, and care for another person” which involves “recognizing the emotions and putting yourself “in the shoes” of others” [4]. Not surprisingly, exploring the phenomenon of empathy is an ongoing research topic. The concept of empathy lacks clarity. It is a vague, multidimensional, and complex phenomenon, and many definitions of empathy have been suggested [8]. There is no single, commonly accepted definition of empathy. In that regard, Sulzer et al. performed a systematic literature review and found that many of the studies that met their search criteria “treated empathy itself as a ‘black box’, using global construct measurements that are unable to shed light on the underlying processes that produce an empathic response” [9]. Furthermore, there is currently scant literature addressing the quality of studies published on this topic in terms of methodology. Different studies conducted using different types of measures produce different results. In 2014, Leombruni et al. stated, “Literature findings show conflicting views in defining and measuring empathy” [10]. Ten years later, the landscape seems to be the same. Moreover, to delve into and gain a deeper understanding of the phenomenon of empathy in undergraduate medical students, we need to delve into its qualitative dimensions using qualitative empirical studies in addition to quantitative empirical studies [11]. However, qualitative studies are scarce [12].

There are three main components of empathy: a cognitive, an affective, and a behavioral. While some definitions of empathy state that it is primarily cognitive, other definitions state that it is both cognitive and emotional [8]. Interestingly, Tsiantou et al. cited literature suggesting that individual differences in emotional or cognitive empathy are rooted in differences in brain structures [8]. Shamay-Tsoory and Lamm “believe that convergent evidence from different techniques will ultimately yield insight into the true mechanisms underlying human empathy, and thus promote basic and clinical research on empathy” [13]. Hojat et al. distinguished between clinical empathy and emotional empathy (which is considered synonymous with sympathy, namely, the emotional reaction to the pain and suffering of the patient). The authors made that distinction mostly because of the different impacts of clinical and emotional empathy on healthcare [14]. Indeed, while an overabundance of clinical empathy (as conceived by Hojat et al.) is beneficial to patient care, an overabundance of emotional empathy may be detrimental to patient care. It can be overwhelming and lead to burnout syndrome. Hojat et al. defined clinical empathy as “a predominantly cognitive (rather than an affective or emotional) attribute (brain mechanism) that involves an understanding (rather than feeling) of the patient’s pain and suffering, combined with a capacity to communicate this understanding (behavioral component), and an intention to help (outcome component)” [14]. 

The Jefferson Scale of Empathy (JSE) is reported to be the first psychometrically sound tool for measuring clinical empathy [14]. Importantly, the JSE focuses mainly on clinical empathy rather than emotional empathy. Studies using the JSE-S have used other instruments to evaluate emotional empathy. The “empathy concern” and “personal distress” dimensions of the Interpersonal Reactivity Index for Medical Students (IRI-MS) have been used to measure emotional (affective) empathy [15]. Interestingly, it is argued that the JSE-S is significantly associated with the cognitive empathy subscales of the IRI [16]. Three versions of the JSE are available: (a) the S version, having as its target population medical students; (b) the HP version, having as its target population a broad spectrum of health professionals, including physicians; and (c) the HPS version, having as its target population all health profession students other than medical students [17]. There are only minor differences between these versions to adjust each one for its particular target population [17]. 

The JSE is widely used by researchers to this day. An “abundance” of evidence (obtained from studies of good methodological quality) supports the construct reliability and validity of the scale. The JSE was translated into various languages and tested in different contexts [17,18,19,20,21,22,23,24,25,26,27,28,29]. Much of this evidence is regarding the JSE-S version of the scale. The JSE has attracted attention from not only the US but also many other countries. “Permission to use the Jefferson Scale of Empathy has been given in 88 countries. The JSE has been translated into 59 languages/dialects (PDF)” [14] and is “the most researched and widely used instrument in medical education research” [17]. It is a sensitive instrument that can detect slight changes in empathy [14]. In that regard, Spatoula et al. suggested that the JSE can detect changes during undergraduate medical education better than other empathy measurement instruments [30]. Note, however, that this study received criticism [31,32].

The psychometric properties of a Greek version of the JSE-S have not been assessed yet in undergraduate medical students. A decade ago, Tsiantou et al. were the first to assess the psychometric properties of the Greek version of the Toronto Composite Empathy Scale (TCES-Gr) in undergraduate dental students. The authors provided evidence to support the reliability and validity of the TCES-Gr [8]. Recently, Voultsos et al. were the first to assess the psychometric properties of the Greek version of the Toronto Composite Empathy Scale (TCES-Gr) in undergraduate medical students [33]. Furthermore, a recently published study confirmed that the Greek version of the JSE-HP has shown good psychometric properties when tested on postgraduate students in different dental specialties and years of study [34]. However, the JSE-S has not been translated into the Greek language and assessed for its psychometric properties before the present study. This was a literature gap that needed to be filled.

To evaluate empathy in the healthcare context it is important to ensure the psychometric rigor of measurement tools. The present study aimed to assess the psychometric properties of the Greek version of the extensively used JSE-S among Greek-speaking medical students. Furthermore, this study used the validated measurement instrument to measure the levels of self-reported empathy among undergraduate medical students in Greece. More precisely, the study aimed at measuring the levels of empathy (mean scores) in three academic years: the first academic year (initial phase of pre-clinical medical education), the fourth academic year (a turning point in which considerable interaction between student and patient begins), and the sixth academic year (the final phase of clinical medical education). These academic years are turning points when studying student empathy throughout their medical studies. As presented below (in the Discussion section), many studies have suggested changes in empathy during medical education, especially in the first clinical academic year. Furthermore, this study sought to identify the association between empathy scores and socio-demographics (potential predictors), including academic year, gender, specialty preferences, and belief in God/supreme power. The association between empathy levels in undergraduate medical students and the factors of academic year, gender, and specialty preferences has been widely explored in prior studies, as presented below (in the Discussion section). Religion can reinforce humanitarian principles and values. Since religion can motivate people to show kindness, compassion, and love to their fellow man, one can assume that there is an association between faith in God and empathy. Interestingly, empathy training can enhance empathy and prevent empathy erosion [4,14,35]. Identifying potential predictors can help instructors create effective empathy-enhancing educational interventions.

## 2. Research Questions

### 2.1. Primary Research Question

This research was directed at answering the following question, which delineates the focus of this study (primary research question):

Is the Jefferson Scale of Empathy—Student version (JSE-S), as translated into Greek by the authors, a reliable and valid tool to measure empathy levels among undergraduate students?

### 2.2. Secondary Research Questions

Furthermore, this research aimed to answer the following secondary research questions:In the validation study, what were the self-reported empathy levels in medical students enrolled in the first year (initial stage of medical studies), fourth year (first clinical academic year), and sixth year (final stage of medical studies)?How were the mean scale scores associated with potential predictors including academic year, gender, specialty preferences, and belief in God/supreme power?

## 3. Methods

### 3.1. Design of the Study

The present study was designed to explore the preliminary psychometric properties of the Greek version of the JSE-S in undergraduate medical students enrolled in the School of Medicine of the Aristotle University of Thessaloniki (AUTH), Greece (the primary aim of the study) and explore the association between empathy levels and socio-demographics (the secondary aim of the study). The study adopted a cross-sectional, comparative–descriptive research design. The English version of the JSE-S was translated into Greek using the back translation method [18]. All students enrolled in the study (N = 213) were invited online (via Google Forms) to complete an anonymous web-based survey including the Greek version of the JSE-S. The data were collected during October and November 2023. The JSE-S was translated into the Greek language and validated after permission was obtained to translate and validate it from the scale developers Asano–Gonnella Center for Research in Medical Education & Health Care (Thomas Jefferson University, Philadelphia, PA, USA).

### 3.2. Test–Retest Reliability

We determined the test–retest reliability with 32 students at the AUTH School of Medicine to further confirm the internal validity of the Greek version of the JSE-S and ensure the representativeness and stability of the outcomes in the medical education sitting over time. The 32 participants in the test–retest reliability study were not among the 213 undergraduate students that participated in this study. They were randomly selected from the 5th year medical students that were attending an undergraduate forensic medicine class during the winter semester of the academic year 2023–2024. Test–retest reliability was evaluated by assessing each participant twice with a 12-day period in between. 

### 3.3. Sample Size

A total of 213 students were recruited from the first, fourth, and sixth academic years of the AUTH Medical School. COSMIN standards (Consensus-based Standards for the Selection of Health Measurement Instruments) require a sample size of ≥100 or seven times the number of the items to evaluate the structural or construct validity or the internal consistency of a scale [36].

### 3.4. The Measurement Instrument

The survey administered to potential participants comprised three parts.

The first part of the survey contained information provided to participants according to the required ethical standards.

The second part of the survey was a questionnaire made up of items focusing on socio-demographics, such as gender, academic year, and career orientation/specialty preference. As the JSE takes about ten minutes to complete, the first and second parts of the questionnaire were structured to take no more than two minutes to complete.

The third part of the survey was the JSE-S translated into Greek. The translation was made in line with the required standards of methodology, as presented below. The JSE-S is a self-administered scale that measures the student’s self-perception of empathy in the healthcare context. The original version of the JSE was developed in 2001 by Hojat et al. [37]. The first version included 90 items. A version of the Delphi method was used to evaluate the psychometric data of the preliminary version of the scale [19]. The current version is a 20-item scale. The items must be answered on a 7-point Likert scale [14]. Each item consists of 7 points ranging from 1 (strongly disagree) to 7 (strongly agree). JSE-S evaluates the following three factors: (a) “Perspective taking” (10 items: 2, 4, 5, 9, 10, 13, 15, 16, 17, and 20), which evaluates the ability to delve into and analyze the patient’s problem “from outside” [38]; (b) “Compassionate care” (8 items: 1, 7, 8, 11, 12, 14, 18, and 19); and (c) “Standing in the patient’s shoes” (2 items: 3 and 6) [20]. Ten out of the total twenty questions are negatively worded and are therefore reverse-scored from 1 (strongly agree) to 7 (strongly disagree) [21]. The total score (the sum of all the item scores) ranges from 20 to 140 [39]. The higher scores are associated with a more empathetic orientation [39].

Establishing cut-off points is essential for identifying medical students with low empathy, with the ultimate goal of implementing curricular interventions to enhance their empathy according to their specific needs, either as a whole or individually [40,41]. Cut-off points may be different in different cultural contexts [41]. Different studies used different cut-off scores for identifying respondents with high, moderate, or low empathy. In a study conducted by Capdevila-Gaudens et al., “scores of over 130 were considered indicative of high empathy (percentile 80)” [39]. Hojat and Gonnella conducted a study and suggested low and high tentative cut-off scores for male students of ≤96 and ≥127, respectively, and low and high scores for female students of ≤102 and ≥129, respectively [42]. Kataoka et al. conducted a study with Japanese medical school matriculants and suggested low and high cut-off scores for male students of ≤91 and ≥126, respectively, and low and high scores for female students of ≤97 and ≥128, respectively [40].

### 3.5. Instrument Translation

We followed the steps required to translate a rating scale [43].

The English (original) version of the JSE-S was independently translated into Greek by two bilingual scientists (one a physician and one a psychologist) who were experienced in the process of scale translation (forward translation). Then, these translators agreed on a final forward translation. To solve possible discrepancies between the original (English) version and the final forward (Greek) version, an expert panel including all members of the research group and external experts (cognitive debriefing) reviewed and compared the two versions. Then, the initial version of the Greek scale was independently back-translated to English by a third translator who did not know the original scale to ensure the conceptual equivalence of the Greek and original English versions [18,43]. Minor modifications were made to the Greek version of the scale to ensure understanding by Greek speakers while achieving equivalency in meaning with the original English version of the JSE-S. Furthermore, to ensure a clear understanding of what we wanted to say in each item, we conducted a pilot study on 15 medical students who were native speakers of Greek. Then, we interviewed them to confirm that they understood the items correctly. None of them identified words or expressions that were hard to understand. The final Greek version of the JSE-S is presented in Appendix A.

### 3.6. Statistical Analysis

Internal consistency reliability: We calculated Cronbach’s α coefficient to examine the internal consistency of the JSE-S. We also calculated Cronbach’s α when a single item was deleted from the 20 items of the JSE-S. An acceptable level of Cronbach’s α is >0.7 [44]. Additionally, we calculated the corrected item–total correlations to examine the degree to which each item of the JSE-S correlated to the total score of the scale. Acceptable values for the item–total correlation are ≥0.30 [45].

Test–retest reliability: We performed test–retest reliability analysis in a subsample of 32 students. In that case, we calculated the intraclass correlation coefficient (ICC) to compare scores on the JSE-S between the two measurements. In particular, we used the two-way mixed-effects model to calculate the ICC. ICC values of <0.5, 0.50 to 0.75, 0.76 to 0.90, and >0.90 are indicative of poor, moderate, good, and excellent reliability, respectively [46].

Construct validity: Using confirmatory factor analysis (CFA), we aimed to ascertain whether the factorial structure of the Greek version of the JSE-S matched that of the original version (i.e., three factors: “perspective taking”, “compassionate care”, “standing in the patient’s shoes”). We used the following goodness-of-fit indices [47,48,49]: the chi-square/degree of freedom (x^2^/df); the goodness-of-fit index (GFI); the normed fit index (NFI); the comparative fit index (CFI); and the root mean square error of approximation (RMSEA). The model in the CFA was considered to have an acceptable or good fit, respectively, when x^2^/df was <5 or <3, the GFI was >0.90 or >0.95, the NFI and CFI were >0.80 or >0.90, and the RMSEA was <0.08 or <0.05.

Known-group validity: We expected differences between the JSE-S score and gender, academic year, specialty preferences, and faith in God. Thus, we performed an independent samples *t*-test to compare gender groups, specialty preferences, and faith in God groups. Also, we used analysis of variance to compare academic year groups. We performed a post-hoc test (Bonferroni test) to compare the three academic year groups.

We used AMOS version 21 (Amos Development Corporation, Meadville, Pennsylvania, USA, 2018) to conduct CFA. All other analyses were conducted with IBM SPSS 21.0 (IBM Corp. Released 2012. IBM SPSS Statistics for Windows, Version 21.0. Armonk, NY, USA: IBM Corp.). We considered *p*-values of <0.05 as statistically significant.

### 3.7. Ethical Considerations

This study was approved by the Bioethics and Deontology Committee of the Faculty of Medicine, Aristotle University of Thessaloniki, Greece (Meeting Number 1/07.11.2023, Prot. 61/2023/13.11.2023). Written informed consent was obtained from all participants. The participants were provided with adequate information regarding the design, purpose, nature, and confidentiality of the study. Before the anonymous questionnaires were completed, the participants were informed that they could decline to participate or withdraw from the study at any time without facing consequences for doing so.

## 4. Results

### 4.1. Participants

In the present study, the research sample consisted of 213 undergraduate medical students. Although a total of 216 students from all three years responded, 213 responses were considered eligible for analysis. Table 1 presents the research sample characteristics. Of the 213 respondents who completed the survey, 63.4% were females and 35.7% were males. Furthermore, 61.6% of students were enrolled in the fourth year, 32.2% were enrolled in the first year, and 6.2% were enrolled in the sixth year. About half of the students (52.9%) reported that they were planning to choose people-oriented specialties. Most of the students reported that they had faith in God/supreme power (72.8%).

### 4.2. Internal Consistency Reliability

The Cronbach’s α for the JSE-S was 0.846. When we deleted items #4 and #19, we found a slight increase in Cronbach’s α (0.847 and 0.854, respectively) (Table 2). Corrected item–total correlations for all items except four items (#3, #4, #18, #19) were above the acceptable value of 0.30 (Table 2). Moreover, Cronbach’s α values for the factors “perspective taking”, “compassionate care”, and “standing in the patient’s shoes” were 0.800, 0.715, and 0.748, respectively. Thus, the internal consistency reliability of the JSE-S was very good.

### 4.3. Test–Retest Reliability

The intraclass correlation coefficient for the JSE-S score in the test–retest study was 0.827 (95% CI = 0.647 to 0.915, *p*-value < 0.001), indicating a high level of reliability.

### 4.4. Construct Validity

The results from the CFA for the Greek version of the JSE-S are shown in Table 3 and Figure 1. The goodness-of-fit indices x^2^/df, RMSEA, and CFI showed a good fit for the model. Moreover, the indices GFI and NFI showed an acceptable fit for the model. Additionally, the correlations between the three factors were positive and statistically significant: 0.81 between the factors “perspective taking” and “compassionate care”; 0.32 between the factors “compassionate care” and “standing in the patient’s shoes”; 0.81 between the factors “perspective taking” and “standing in the patient’s shoes” (*p* < 0.001 in all cases). Furthermore, standardized regression weights between the 20 items and the three factors ranged from 0.197 to 0.899. Therefore, the Greek version of the JSE-S confirmed the three-factor structure of the original version of the scale.

### 4.5. Known-Group Validity

In the present study, the participants (undergraduate medical students) showed moderate empathy levels across the first, fourth, and sixth academic years, according to the cut-offs presented above [22,39,42]. Females scored higher than males on the Greek version of the JSE-S (108.8 vs. 102.8, *p*-value < 0.001). Moreover, students enrolled in the fourth academic year showed higher empathy mean scores than those enrolled in the first year (108.6 vs. 103.6, *p*-value = 0.012). Sixth-year students showed empathy levels similar to those of the first-year students. We did not find statistically significant differences in the empathy mean scores by specialty preferences or faith In God/supreme power. Detailed results from the known-group validity analysis are shown in Table 4.

## 5. Discussion

The present study provided satisfactory evidence that the Greek version of the JSE-S is a psychometrically sound tool to assess the levels of empathy among undergraduate medical students.

### 5.1. Association of Empathy with Academic Year

In the present study, the participants showed moderate empathy levels across the first, fourth, and sixth academic years. Last year, Voultsos et al., using the TCES-Gr, conducted a study similar to the present study in the same educational setting. The authors found that “the mean total score of empathy showed that students had a moderately high empathy” [33]. Overall, in the present study, we observed almost similar empathy levels across the three explored academic years (first, fourth, and sixth), with fourth-year students showing a slight (even though statistically significant) increase in empathy levels as compared with first-year and sixth-year students. This was inconsistent with the vast majority of prior studies. It may be partially attributed to the fact that fourth-year students were attending the main Medical Ethics course of their undergraduate medical education at the time of the survey. However, our findings were consistent with a minority of previous studies. It is extensively suggested in the currently available literature that empathy changes (often undergoing erosion) during undergraduate medical education, with most studies suggesting that the decline in empathy starts in the third (or fourth) year of studies; namely, in the transition from the final pre-clinical phase to the clinical phase of medical training [3,14,28,33,50,51,52,53,54,55,56,57,58,59,60,61,62,63,64,65,66,67,68,69]. Almost all the published studies used the JSE. This effect was first noticed by Hojat et al., using the JSE in 2004 [70] and then in 2009 (in longitudinal studies) [3].

Some studies report nuances when suggesting the effect of empathy decline during undergraduate medical education. Akgün et al. found that while clinical empathy levels decreased significantly after the third academic year, “the general empathy levels decreased less” [61]. Baig et al. state that empathy declines as medical education progresses through the years, with a slight increase in empathy levels being seen in the final year [62]. A study conducted two years ago at the School of Medicine of the Aristotle University of Thessaloniki using not the JSE-S but the TCES-Gr found a statistically significant empathy decline among male students throughout their medical studies, with female students demonstrating no statistically significant empathy decline [33]. A similar gender difference in empathy decline was reported by Lwow et al. [71].

However, available evidence supporting the effect of empathy decline during undergraduate medical education appears inconclusive. Colliver et al. conducted a literature review and found that “the evidence does not warrant the strong, disturbing conclusion that empathy declines during medical education” [72]. Costa et al. stated, “a latent growth model suggests that empathy of medical students does not decline over time” [73].

Overall, however, there is no agreement on the trend in changes in the levels of empathy among medical students [59]. Ferreira-Valente et al. conducted a scoping review and concluded, “At the present moment, the literature does not offer clear conclusions relative to changes in student empathy throughout medical school” [74]. Calzadilla-Núñez et al. expressed it best in saying that the effect of empathy decline is a “specific element of several different models of empathic response (and of its components)”: (a) decline, (b) stable empathy over time, and (c) constant increase in empathy throughout the full medical course [75].

Some studies suggest that students’ empathy increases throughout the course of their medical studies [22,76,77,78]. Other studies found no significant empathy decline throughout medical studies [12,41,79,80,81,82,83,84,85,86]. Other studies found no statistically significant changes in empathy across academic years [41,81,87]. Kötter et al. stated, “In our study, empathy scores were shown to be relatively stable during medical education with a tendency to increase” [81]. Note that Blanco et al. found no empathy changes by academic year, but they noticed a slight increase in cognitive empathy among female medical students [41]. Adatia et al. found that mean empathy scores increased after third year (mean empathy score in Med 3, 107.5 and in Med 4, 115.3). However, this increase was not statistically significant [53]. Most but not all of the abovementioned studies used the JSE. Interestingly, while the present study is in line with the studies mentioned above, it was conducted using the Greek version of the JSE-S.

Moreover, empathy may fluctuate over the course of undergraduate medical studies. Wahjudi et al. found that students’ empathy increased in the first academic years, but “declined significantly upon entering the first clinical year”, and then increased [88]. An Indian study found that empathy scores tend “to decline initially and then rebound over time” [89].

The widely suggested effect of empathy decline over the course of medical studies may be affected by methodological inconsistencies. As most studies conducted worldwide used the JSE, one might hypothesize that the suggestion that empathy “declines” throughout medical education may be partly true due to the extensive use of the JSE in the studies. For instance, Quince et al. used the IRI and found no empathy decline [83]. Note, however, that Voultsos et al. using the TCES found a statistically significant decline among male medical students [33]. On the other hand, a Japanese study conducted by Kataoka et al. using the JSE found that “empathy scores increased as students progressed through medical school” [22]. Furthermore, many of the abovementioned studies that found stable empathy distribution used the JSE. Moreover, the suggestion that empathy “declines” throughout medical education may be partly because longitudinal studies are scarce and there is a theoretical confusion regarding the multidimensional concept of empathy [54,65,73,90,91]. Qualitative empirical studies investigating how “students themselves conceive and understand empathy” during the course of studies would contribute to further exploring the phenomenon of the decline in empathy [54]. Indeed, it is a paradox that student empathy decreases during the clinical academic years; namely, a stage of medical education that requires higher levels of empathy. In that regard, Laughey et al. stated “The problems of empathic burden, negative role modelling and of finding empathy difficult for challenging patients may account for some of the empathy decline” [92]. In addition, Hizomi et al. attempted to explain the effect of empathy decline by citing the long educational programs, insufficient bedside interactions, and students’ negative experiences and stressful perspectives on medical education [59]. Furthermore, the pattern of empathy reaction during medical studies may be profoundly affected by socio-cultural factors. Ponnamperuma et al. conducted a literature review and concluded that the patterns of empathy change during medical education are similar “within approximate geo-sociocultural clusters” [2]. In that regard, Sng et al. stated, “While the empathy construct in multicultural Singapore is congruent with a Western model, important differences remain” [68]. At any rate, the so-called “hidden curriculum” may be one of the main reasons for the empathy decline during the period of undergraduate medical education [65,93]. Peng et al. conducted a study in North America and concluded that “cynicism develops progressively and is compounded by conflicts between the hidden and formal curriculum” [93]. At any rate, factors for the decline in empathy need to be further explored.

### 5.2. Association of Empathy with Gender

In the present study, female students showed statistically significantly higher empathy levels than male students. This finding is consistent with the vast majority of prior studies. Many studies conducted across various cultures reported statistically significant differences in overall empathy scores between male and female students and confirmed that female medical students are more empathetic than their male counterparts [10,14,15,16,17,25,29,33,37,55,57,58,60,62,63,64,66,67,68,69,76,83,84,89,94,95,96,97,98,99,100,101]. As mentioned above, this discrepancy changes across the academic years. Given that the vast majority of the abovementioned studies used the JSE-S, one might hypothesize that the discrepancy in empathy by gender is due to the use of that measurement instrument. Note, however, that there are studies conducted using the JSE-S that found no statistically significant differences in the empathy mean scores by gender among undergraduate medical students [26,28,53,59,82,84,87,102]. On the other hand, Javaeed et al. concluded that “Female students showed higher empathy than male ones for some IRI scale items” [103]. Ursoniu et al. used the Toronto Empathy Questionnaire (TEQ) and found significantly higher empathy scores for female medical students compared to male medical students in convergent validity [104].

Genetic predisposition and social learning are among the reasons provided as explanations for gender differences in favor of women among undergraduate medical students [18]. Rueckert and Naybar suggested that gender differences in empathy might be associated with the activation of the right cerebral hemisphere [105]. Christov-Moore and Iacoboni concluded that the findings of their study “suggest that localized internal somatomotor representations of others’ pain, a functional index of bottom-up resonance processes, are stronger in female subjects” [106]. Note, however, that Baez et al. suggested that gender differences in empathy “are highly driven by the assessment measure” [107]. Furthermore, Ardenghi et al. concluded that “the attachment styles of medical students are related to self-evaluated empathy, over and above the effects of gender and age differences” [108].

### 5.3. Association of Empathy with Specialty Preferences

In the present study, we did not find an association between empathy levels and students’ specialty preferences. This finding is in line with the minority of prior studies. Some studies confirm that undergraduate medical students who are inclined to choose “specialties that require continuous and prolonged encounters with patients” (“people-oriented” specialties, such as pediatrics, family medicine, general internal medicine, and psychiatry) receive significantly higher empathy scores than those inclined to choose specialties that do not require continuous and prolonged encounters with patients (“technology/procedure-oriented” specialties, such as pathology, radiology, anesthesiology, and surgical specialties) [14,28,37,41,50,58,60,64,67,68,76,81,84,85,101,109]. Luna et al. combined gender and career preference and concluded: “men, preference for technology-oriented specialty, less empathy. Women, preference for patient-oriented specialty, greater empathy” [78]. Yeo et al. combined academic year and gender and concluded that “students who preferred person-oriented specialties had higher empathy scores when they entered their fourth academic year compared to objectively-oriented students” [110].

Notwithstanding, studies found no statistically significant difference in empathy scores by specialty preference [53,66,79,87,102]. The present study is in line with these studies.

Tiwari et al. found higher empathy scores among medical “students having previous experience of hospital stay as an attendant of a patient” [101].

Ultimately and most importantly, it should be highlighted that Andersen et al. performed a systematic review that revealed equivocal evidence on how the empathy level among medical students is associated with academic year, gender, and specialty preference [111].

### 5.4. Religion/Spirituality

Furthermore, we did not find an association between empathy levels and faith in God/supreme power. Such an association has not been widely suggested in the currently available academic literature. Damiano et al. conducted a study with Brazilian medical students and, among other things, they concluded that religiosity is one of the significant independent predictors of empathy [112]. Málaga et al. found that religious beliefs are related to higher levels of empathy (*p* = 0.0001) [55]. In a study conducted by Hamidia et al., “Pearson’s correlation coefficient revealed a strong positive significant relationship between student’s perspective to clinical empathy and spiritual well-being (r = 0.56)” [113]. About ten years ago, Pawlikowski et al. conducted a study and concluded that religiosity is positively associated with empathy (r = 0.20; *p* < 0.01). The authors stated “Physicians’ religiosity is an essential factor shaping their attitude towards patients” [114].

Ultimately, as Hojat et al. did not find any relationship between clinical empathy and health professional students’ ages, we did not explore the association between clinical empathy and age [14].

### 5.5. Empathy Levels

A large number of previous studies which come from different cultural environments report moderate overall empathy levels in undergraduate medical students. In these studies, the mean empathy scores range roughly from about 90 to 120, with most of them ranging from about 100 to 115. Studies conducted in Spain and Peru have shown high empathy scores (ranging roughly from 115 to 120), with studies conducted in India and Pakistan having shown low empathy scores (ranging roughly from 90 to 110). Below, we present a literature review. 

Neeraj et al. conducted a study in India and found that the mean empathy score was 100.75 [1]. Leombruni et al. (in Italy) found that “The mean total score of the JSE S-Version was 108.71 (SD = 10.60)” [10]. Capdevila-Gaudens et al. (in Spain) found that the mean numerical value for empathy “was 120.6 ± 11.8 points (maximum 140), with 18.8% of participants having high levels of empathy (>130)” [39]. Benabbas (in Iran) found that “the mean score (SD) of empathy was 101 (15.6)” [115]. Kataoka et al. (in Japan) found that the mean empathy score was 104.3 (SD = 13.1) [22]. Ferreira-Valente et al. (in Spain) assessed the psychometric properties of the Spanish version of the JSE-S and found total empathy score 117.56 (SD = 10.43) [24]. Sheriat et al. (in Iran) found that the mean score of JSE in the sample was 101.4 (SD = 14.5) [25]. Paro et al. (in Brazil) assessed the psychometric properties of the Brazilian version of the JSE-S and found that “the mean total score for the JSE was 114.95 (SD = 12.41)” [26]. Blanco et al. (in Spain) found that “The overall mean JSE score was 115.22 points (SD = 14.24) for men and 121.17 points (SD = 11.18) for women” [41]. Málaga et al. (in Peru) found that “The mean score was 118.01 (range: 20 to 140), standard deviation, 11.61; median, 120” [55]. Iqbel et al. (in Saudi Arabia) found that the overall mean empathy score was 105.18 ± 12.51 [57]. Hizomi et al. (in Iran) found that the overall mean empathy score was 110.19 ± 13.61 and 103.52 ± 20.00 in first- and last-year medical students, respectively [59]. Mirani et al. (in Pakistan) found that “The mean ± SD empathy score of all students was 98.11 ± 12.31 (range: 20–140)” [60]. Park et al. (in Korea) “analyzed 5343 questionnaires and found a mean empathy score of 105.9 ± 12.8” [63]. Shashikumar et al. (in India) found that “The mean empathy score of all the students was 102.91” and notice that it is “much lower than that reported by Chen et al. (114.3) but similar to those reported by Katoaka et al. (104.30) and Rahimi et al. (104.1)” [66]. Raof and Yassin (in Iraq/Kurdish medical students) found that “the mean empathy score was 101.9 ± 19.2” [67]. Sng et al. (in Singapore) found that “The overall mean was 112.18 (standard deviation, 11.381)” [68]. Ulloque et al. (Argentina) found a mean empathy score 112.84 (SD=11.581) [77]. Kötter et al. (in Germany) in cross-sectional samples found JSE-S sum scores ranging from 113.93 (SD = 0.6) to 117.53 (SD = 10.3) [81]. Samarasekera et al. (in Singapore) found that in their study “average scores for both cohorts were 113.94 and 115.66” [86]. Chatterjee et al. (in India) conducted a study in India and found state, “The mean empathy score in our study was 96.01 out of a maximum of 140, with a standard deviation of 14.56” [89]. Tiwari (in India) conducted a study states, “The arithmetic mean (±SD) of empathy scores was 99.87 ± 14.71” [101]. Wen et al. (in China) found that the mean empathy score was 109.60 (SD = 12.09) [94]. Mostafa et al. (in Bangladesh) found that “The mean empathy score was 110.41 ± 13.59” [95]. Archer and Turner (in South Africa) found that “The mean JSE was 109.98 (SD = 12.54), which is lower than most internationally reported scores” [96]. Yi et al. (in China) found that “The total mean was 104.66 (SD = 13.09), and scores ranged from 39 to 137” [97]. Shaheen et al. (in Pakistan) found that “overall mean empathy level was 90.63 ± 11.55” [98]. Nasiri et al. (in Iran) “the total mean score of empathy was 98.15 ± 13.29” [99]. Assing Hvidt et al. (in Denmark) found that “The overall mean score was 112.7” (SD = 10.8) [109].

### 5.6. Cronbach’s Alpha Values

Furthermore, below we present a literature review on the Cronbach’s alpha values reported in studies that assessed the JSE-S in different cultural contexts. The values range roughly from 0.7 to 0.9:

Ghardallou et al. (in Tunisia) assessed the psychometric properties of a French version of the JSE-S and found that “Reliability analysis reported an acceptable level of internal consistency, with an overall Cronbach’s alpha of 0.78 (95% CI [0.75, 0.80]) [38]. Jiang et al. (in China) assessed the reliability and validity of the Chinese version of the JSE-S and found a Cronbach alpha of 0.861 [23]. Ferreira-Valente et al. (in Spain) assessed the psychometric properties of the Spanish version of the JSE-S and found an overall JSE-S Cronbach alpha of 0.78 [24]. Sheriat et al. (in Iran) found an overall JSE-S Cronbach alpha of 0.79 [25]. Paro et al. (in Brazil) assessed the psychometric properties of the Brazilian version of the JSE-S and found “Cronbach’s coefficient for the scale was 0.84” [26]. Málaga et al. (in Peru) found “a Cronbach’s alpha coefficient of 0.76” [55]. Mirani et al. (in Pakistan) found that “the internal consistency of JSE-S was 0.71 [60]. Magalhães et al. (Portugal) found that “Cronbach’s Alpha for total scale was 0.77 which is similar to previous reliability values (0.76) reported in the Portuguese publication” [76]. Ulloque et al. (Argentina) Cronbach’s alpha values were satisfactory (non-typified = 0.766; typified = 0.789) “and this suggested internal reliability” [77]. Chatterjee et al. (in India) state, “The 20-item empathy scale was observed to have good internal consistency in this population group (α = 0.765)” [89]. Wen et al. (in China) found that the Cronbach’s alpha coefficient was 0.83 [94]. Mostafa et al. (in Bangladesh) found that “Cronbach’s alpha coefficient was 0.88” [95]. Archer and Turner (in South Africa) found that “The Cronbach’s alpha coefficient was 0.81” [96]. Yi et al. (in China) found that Cronbach’s alpha their study was 0.82 [97]. Jung et al. (in Korea) found that “the reliability of the Korean JSE-S was 0.910 by Cronbach α coefficient” [102].

## 6. Implications

This study provides preliminary evidence supporting the use of the Greek version of the JSE-S in undergraduate Greek-speaking medical students. The instrument can contribute to evaluating the overall empathy of students in medical education settings, identifying the students’ individual differences and empathic profiles, conducting longitudinal studies, and identifying predictors of student empathy; namely, factors to be considered when developing and implementing interventions to enhance the students’ empathy and prevent or mitigate the erosion (if any) of empathy over academic years, according to the specific needs of students. To further explore the phenomenon of empathy among undergraduate medical students, longitudinal cross-sectional and qualitative empirical research is required. The Greek version of the JSE-S can be used to obtain empathy scores when conducting longitudinal cross-sectional and mixed quantitative and qualitative empirical research.

There is no agreement on the best intervention for enhancing the levels of empathy in undergraduate medical students. Many different interventions have been proposed and tested by researchers. Many years ago, Hojat briefly described ten approaches for enhancing empathy: “improving interpersonal skills, audio- or video-taping of encounters with patients, exposure to role models, role playing (aging game), shadowing a patient (patient navigator), hospitalization experiences, studying literature and the arts, improving narrative skills, theatrical performances, and the Balint method” [3]. Valuing non-verbal compassionate communication, reflection, and enhancing students’ self-esteem seem to be helpful [52,92,116]. However, it is not easy to determine the most effective empathy-enhancing intervention. Various studies on the effectiveness of the proposed interventions have produced mixed results. Further longitudinal empirical research might be helpful. Enhancing the levels of empathy in undergraduate medical students should be integrated into the medical curriculum. Physician empathy is a significant indicator of healthcare service quality [117]. 

## 7. Limitations and Strengths

The present study is the first to explore the psychometric properties of the Greek version of the JSE-S. When we asked permission from the Asano–Gonnella Center for Research in Medical Education & Health Care (Thomas Jefferson University, USA), they informed us that they did not have a Greek translation of the S version of the JSE before the present study. Another strength of the study is that we conducted a test–retest reliability with a sufficient number of participants to further confirm the reliability of the tested scale. However, some limitations of the study cannot be excluded. The sample size was small. Furthermore, “acquiescence response style”, “confirmation bias”, and “careless responding” cannot be excluded and (if present) may lead to invalid responses [21]. In addition, we should bear in mind that the JSE-S is a self-report instrument and as such may not indicate actual student attitudes toward empathy. Importantly, self-report assessment and an unclear definition of empathy might be considered limitations of a study of student empathy using a self-report scale. It is argued that there is a “discrepancy between self-administered empathy scores and observed empathic behaviours” [118]. “Physician empathy assessed by the self-reported scale is inversely associated with patient perceptions” [119]. In that regard, it is noted that when students are invited to participate actively and voluntarily, they might express a positive attitude toward empathy to be in agreement with the spirit of the medical school curriculum (official or hidden). Furthermore, a study of empathy in medical students may not indicate actual student attitudes toward empathy for another reason. Costa-Drolon et al. conducted a systematic review and suggested that “medical students are beset by theoretical confusion regarding the concept of empathy” [54].

Moreover, the use of the JSE in a study of empathy may imply a limitation of the study. Williams and Beovich emphasized that while “the JSE demonstrates robust structural validity, internal consistency, and convergent validity”, “current evidence is limited for the properties of reliability, measurement error, and cross-cultural validity” [19]. Therefore, the authors state that “a degree of caution should be considered in drawing conclusions when using the JSE with regard to these properties” [19]. In this regard, it is argued that the suggestion that empathy “declines” throughout medical education may be due in part to the extensive use of the JSE [90]. However, the present study does not provide evidence to support this assumption.

## 8. Conclusions

The present study provided satisfactory evidence that the Greek version of the JSE-S is a psychometrically sound, reliable, and valid measurement instrument to assess the levels of empathy among undergraduate medical students. The participants showed moderate empathy levels across the first, fourth, and sixth academic years. Not surprisingly, female students showed statistically significantly higher empathy levels than male students. Interestingly, first- and sixth-year students showed similar empathy levels, with fourth-year students showing statistically significantly higher (though not considerably higher) empathy levels. Furthermore, we did not find statistically significant differences in empathy levels by specialty preferences or faith in God/supreme power. The use of the scale can contribute to designing medical curricula aimed at enhancing empathy in medical students and ultimately improving the quality of medical care. However, the association of empathy levels with academic year and specialty preferences should be further investigated.

## Figures and Tables

**Figure 1 behavsci-14-00195-f001:**
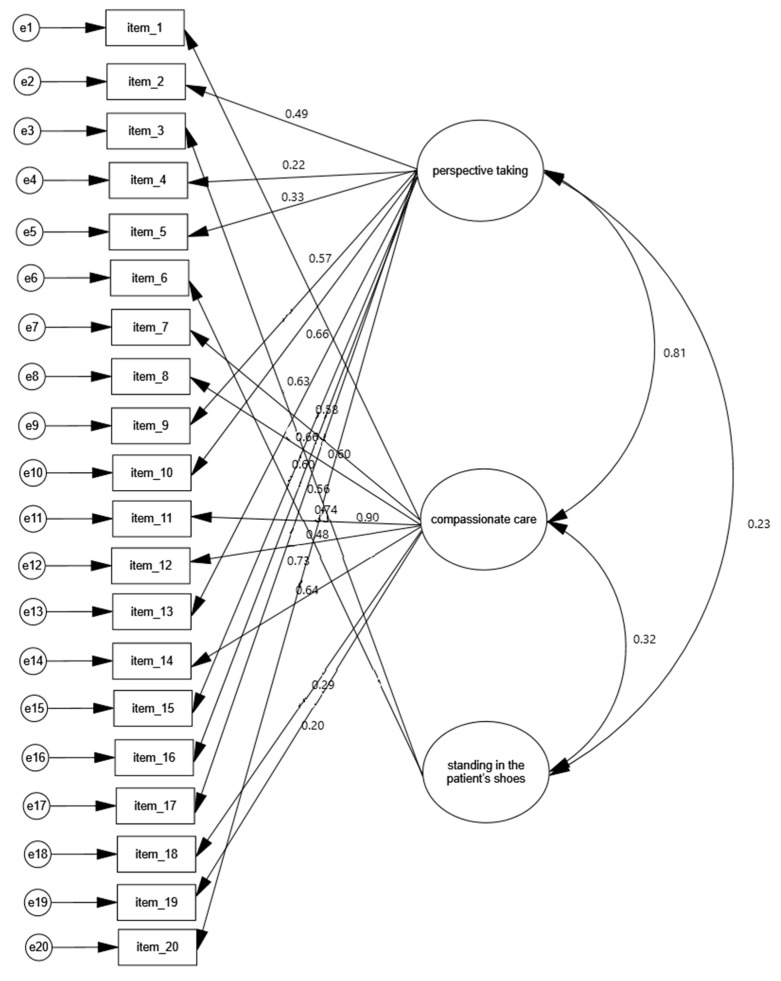
Confirmatory factor analysis of the Greek version of the JSE-S and regression/correlation values.

**Table 1 behavsci-14-00195-t001:** Sample characteristics.

Characteristics	N	%
Gender (n = 211)		
Males	76	35.7
Females	135	63.4
Academic year (n = 211)		
First	68	32.2
Fourth	130	61.6
Sixth	13	6.2
People-oriented specialties preference (n = 208)		
No	98	47.1
Yes	110	52.9
Faith in God (n = 202)		
No	55	27.2
Yes	147	72.8

**Table 2 behavsci-14-00195-t002:** Corrected item–total correlations and Cronbach’s α when a single item was deleted from the 20 items of the JSE-S.

Items	Corrected Item–Total Correlation	Cronbach’s Alpha If Item Deleted
1	0.514	0.835
2	0.433	0.841
3	0.275	0.846
4	0.196	0.847
5	0.314	0.845
6	0.300	0.846
7	0.494	0.837
8	0.476	0.837
9	0.537	0.835
10	0.577	0.833
11	0.648	0.831
12	0.414	0.840
13	0.522	0.838
14	0.543	0.834
15	0.563	0.833
16	0.503	0.838
17	0.489	0.837
18	0.280	0.846
19	0.207	0.854
20	0.629	0.833

**Table 3 behavsci-14-00195-t003:** Confirmatory factor analysis for the Greek version of the JSE-S.

Model	x^2^	df	x^2^/df	RMSEA	GFI	NFI	CFI
20 items with three-factor structure	157.994	149	1.060	0.017	0.931	0.883	0.992

**Table 4 behavsci-14-00195-t004:** Mean scores of the JSE-S by gender, academic year, specialty preferences, and faith in God.

Characteristics	Mean	Standard Deviation	*p*-Value
Gender			<0.001 ^a^
Males	102.8	12.6	
Females	108.8	10.8	
Academic year			0.008 ^b^
First	103.6	11.5	
Fourth	108.6	11.8	
Sixth	103.0	8.5	
People-oriented specialties preference			0.133 ^a^
No	105.4	12.5	
Yes	107.8	11.0	
Faith in God			0.121 ^a^
No	109.1	11.7	
Yes	106.1	11.8	

^a^ independent samples *t*-test. ^b^ analysis of variance.

## Data Availability

Data are available upon request to the first author.

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
