# Peer review of "The Greek Jefferson Scale of Empathy—Medical Student Version (JSE-S): Psychometric Properties and Its Associated Factors"

_behavsci, 2024, doi:10.3390/bs14030195_

Round 1
Reviewer 1 Report
Comments and Suggestions for Authors
The objective of the study was fulfilled and the introduction was well-written. Elaborate discussion was presented. I have some suggestions:
Line 108 - "We determined the test–retest reliability with 32 students at the AUTH School of Medicine to further confirm the internal validity of the Greek version of the JSE-S and ensure the representativeness and stability of the outcomes in the medical education sitting over time".
- Were these 32 students among the 213 students? Clarify.
Line 118 - The measurement instrument
- Too much details. Reduce some details to be interesting to the readers.
- In conclusion part - lines 630-634 - you don't need to mention here in the conclusion comparing your results with other studies. Just mention your conclusion what you observed from your study.
Reviewer 2 Report
Comments and Suggestions for Authors
Review of work
Title: Psychometric Properties of the Greek Jefferson Scale of Empathy—Student Version among Undergraduate Medical Students and its Association with Potential Predictors
Journal: Behavioral Sciences
Congratulations to the authors for their efforts and the critical findings they reached
Title: too long to shorten
Abstract: Rewrite. Numbers are unnecessary in the abstract. Please write the abstract again so that you do not speak in statistical terms but present the findings about their significance.
Introduction
Line 56/57 – sentence without reference, grammatically incorrect; move it somewhere where it fits meaningfully
List theoretical and scientific dilemmas as well as scientific studies organised around them. The topic has been widely discussed. Numerous psychometric analyses of the questionnaire have been done. There are many closed and open scientific and theoretical questions. This is not evident from your theoretical work.
Test/retest - it is necessary to describe how students were selected for the test and retest
The instrument used is nicely defined in the method part, the instruments section, but the text from lines 144 to 220 does not belong in the method; it should be in the theoretical part.
Part of the discussion and results are excellently written.
​
Reviewer 3 Report
Comments and Suggestions for Authors
The structure of the manuscript is unusual regarding the theoretical input during the method section. The introduction is rather short and missed a definition of empathy as well as the actual research status about the effect of empathy for the patient, changes in empathy during medical education (or age?) as well as possible influencing factors. Especially the association between 'faith in god' and empathy needs some theoretical background.
Some of this content could be find in 3.4 the measurement instruments, what seems not the right place.
The aim of 3.4.1 regarding the present manuscript is not clear.
The discussion is transparent and readable, nevertheless relative long. Maybe the authors could be more 'brought to the point' in some parts.
Round 2
Reviewer 2 Report
Comments and Suggestions for Authors
The authors significantly improved the work and responded to all comments. Congratulations